



# Geometric estimation of volcanic eruption column height from GOES-R near-limb imagery – Part 1: Methodology

Ákos Horváth[1], James L. Carr[2], Olga A. Girina[3], Dong L. Wu[4], Alexey A. Bril[5], Alexey A. Mazurov[5], Dmitry V. Melnikov[2], Gholam Ali Hoshyaripour[6], Stefan A. Buehler[1]

[1]Meteorological Institute, Universität Hamburg, Hamburg, Germany
[2]Carr Astronautics, Greenbelt, MD, USA
[3]Institute of Volcanology and Seismology, Far East Branch of the Russian Academy of Sciences (IVS FEB RAS), Petropavlovsk-Kamchatsky, Russia
[4]NASA Goddard Space Flight Center, Greenbelt, MD, USA
[5]Space Research Institute of the Russian Academy of Sciences (SRI RAS), Moscow, Russia
[6]Institute of Meteorology and Climate Research, Karlsruhe Institute of Technology (KIT), Karlsruhe, Germany

*Correspondence to*: Ákos Horváth (akos.horvath@uni-hamburg.de, hfakos@gmail.com)

**Abstract.** A geometric technique is introduced to estimate the height of volcanic eruption columns using the generally discarded near-limb portion of geostationary imagery. Such oblique observations facilitate a height-by-angle estimation
method by offering close to orthogonal side views of eruption columns protruding from the Earth ellipsoid. Coverage is restricted to daytime point estimates in the immediate vicinity of the vent, which nevertheless can provide complementary constraints on source conditions for the modelling of near-field plume evolution. The technique is best suited to strong eruption columns with minimal tilting in the radial direction. For weak eruptions with severely bent plumes or eruptions with expanded umbrella clouds the radial tilt/expansion has to be corrected for either visually or using ancillary wind profiles.
Validation on a large set of mountain peaks indicates a typical height uncertainty of ±500 m for near-vertical eruption columns, which compares favorably with the accuracy of the common temperature method.

## 1 Introduction

Volcanic eruptions pose significant hazards to aviation, public health, and the environment (Martí and Ernst, 2005). Risk assessment and mitigation of these hazards is supported by atmospheric dispersion models, which require the eruptive source
parameters, especially plume height and the mass eruption rate (MER), as key inputs (Peterson et al., 2015). Plume height and MER are related by dynamics and the latter scales approximately as the fourth power of the former. Thus, a small error in plume height leads to a large error in MER, estimates of which can consequently have a factor of 10 uncertainty (Bonadonna et al., 2015). The mass eruption rate is commonly estimated from plume height observations using semi-empirical relationships, Sparks-Mastin curves, derived from buoyant plume theory and historical eruption data (Mastin et al.,
2009; Sparks et al., 1997). An alternative is to use simplified 1D cross-section-averaged (Folch et al., 2016) or 2D Gaussian (Volentik et al., 2010) plume-rise models, which can be inverted efficiently to estimate MER from plume height.



Many techniques have been developed over the years to measure volcanic plume height (for a comparative overview see Dean and Dehn, 2015; Merucci et al., 2016; Zakšek et al., 2013; and references therein). Ground-based methods rely on weather radars, lidars, or video surveillance cameras. Space-based methods include radar and lidar observations, radio occultation, backward trajectory modeling, geometric estimates from shadow length and stereoscopy, as well as radiometric estimates utilizing the $CO_2$ and $O_2$ absorption bands and infrared (IR) channels.

The height retrieval technique offering the best spatial and temporal coverage globally is the spaceborne 'temperature method', which is based on IR brightness temperatures (BTs) routinely available from a large suite of imaging radiometers onboard both polar orbiter and geostationary satellites. In its simplest and still oft used single-channel form, the method determines plume height by matching the 11 μm BT to a temperature profile from a numerical forecast, assuming an opaque plume in thermal equilibrium with its environment. Both of these assumptions, however, can be invalid.

The plume tops of the largest explosions, especially ones that penetrate the stratosphere, might be in thermal disequilibrium due to decompression cooling in a stably stratified atmosphere. Undercooling can lead to a cloud top that is tens of degrees colder than the minimum temperature of the surrounding ambient, in which case the satellite measured BT cannot be converted to height (Woods and Self, 1992). Thermal disequilibrium of the opposite sign might also occur because the increased absorption of solar and thermal radiation by volcanic ash can cause significant local heating (Muser et al., 2020), resulting in negatively biased height retrievals.

A more common problem is that the plume, especially its dispersed part further from the vent, is semi-transparent to IR radiation and deviates strongly from blackbody behavior; hence, the 11 μm BT is warmer than the effective radiative temperature. Surface contribution to the measured BT leads to underestimated plume heights (Ekstrand et al., 2013). This low bias can be somewhat reduced by using only the minimum (dark pixel) BT of the plume least affected by surface radiation.

A more sophisticated treatment of semitransparency effects, however, requires BTs from multiple IR channels. Pavolonis et al. (2013) developed a volcanic ash retrieval based on the 11 μm and 12 μm split-window channels and the 13.3 μm $CO_2$ absorption band, the latter providing the needed height sensitivity for optically thin mid- and high-level plumes. The algorithm solves for the radiative temperature, emissivity, and a microphysical parameter of the plume by optimal estimation. These parameters are then used to determine plume height, effective ash particle radius, and mass loading for four different mineral types (andesite, rhyolite, gypsum, and kaolinite) to account for uncertainty in chemical composition.

All brightness temperature-based height retrievals are however problematic near the tropopause due to the characteristic temperature inversion. Small lapse rates and nonunique solutions lead to a significantly increased height uncertainty. Over- and underestimation are both possible depending on whether the forecast temperature profile is searched top-to-bottom or bottom-to-top for a matching height. As a result of the listed error sources (emissivity, chemical composition, lapse rate), temperature methods have typical absolute uncertainties of 1–2 km for low- and mid-level (<7 km) plumes and 3–4 km for high-level (>7 km) plumes compared to lidar or stereo heights generally considered the most accurate (Flower and Kahn, 2017; Pavolonis et al., 2013; Thomas and Siddans, 2019).



Globally applicable near-real time techniques, such as the satellite temperature method, are nevertheless indispensable to support operations at volcanic ash advisory centers and mitigate aviation and health hazards. The single-channel temperature method is part of the VolSatView information system (Bril et al., 2019; Girina et al., 2018; Gordeev et al. 2016) operated by the Kamchatka Volcanic Eruption Response Team (KVERT; Girina and Gordeev, 2007). The multichannel retrievals form
the core of the VOLcanic Cloud Analysis Toolkit (VOLCAT) and are produced from Geostationary Operational Environmental Satellite-R (GOES-R) and Himawari-8 imager data by the National Oceanic and Atmospheric Administration (NOAA) and the Japanese Meteorological Agency (JMA). The pursuit of new techniques is still worthwhile though, given the large uncertainty of existing retrieval algorithms (von Savigny et al., 2020).

Our proof-of-concept study introduces a simple geometric technique to derive *point estimates* of eruption column height
in the vicinity of the vent from side views of the plume captured in near-limb geostationary images. In planetary science, topography is often estimated by the radial residuals to a best-fit ellipsoid along a limb profile. Such 'limb topography' was derived for Io (Thomas et al., 1998), saturnian icy satellites (Nimmo et al., 2010; Thomas, 2010), and Mercury (Oberst et al., 2011), to mention a few. Limb images were also used to estimate the height of ice and dust clouds on Mars (Hernández-Bernal et al., 2019; Sánchez-Lavega et al., 2015, 2018) and even the height of volcanic plumes on Io (Geissler and
McMillan, 2008; Spencer et al., 2007; Strom et al., 1979). Closer to home, near-limb images from geostationary satellites were used to reconstruct the atmospheric trajectory of the 2013 Chelyabinsk meteor (Miller et al., 2013) and study the altitude of polar mesospheric clouds (Gadsden, 2000a, b, 2001; Proud, 2015; Tsuda et al., 2018). Apart from these two applications, however, the near-limb portion of geostationary images is completely unused for any quantitative geophysical analysis.

Here we exploit exactly these oblique observations, which provide side views at almost right angle of volcanic eruption columns protruding from the Earth ellipsoid. The proposed height-by-angle technique is based on the finest resolution daytime visible channel images and it is analogous to the astronomical height retrievals and the height estimation methods from calibrated ground-based video camera footage (Scollo et al., 2014). In particular, we take advantage of the current best-in-class Advanced Baseline Imager (ABI) onboard the GOES-R satellites, which offers nominally 500-m resolution red band
imagery every 10 minutes combined with excellent georegistration (Schmit et al., 2017). Part 1 of the paper describes plume height estimation specifically from ABI data, but the method is equally applicable to data from the nearly identical Advanced Himawari Imager (AHI) onboard the Himawari 3rd generation satellites. Part 2 of the paper then demonstrates the technique for seven recent volcanic eruptions observed by GOES-17 in Kamchatka, the Kuril Islands, and Papua New Guinea, including the 2019 Raikoke eruption, and compares the side view plume height estimates with those from the IR
temperature method, stereoscopy as well as ground-based video camera and quadcopter images.



## 2 ABI and AHI limb observations

### 2.1 Full disk fixed grid image

We use the highest resolution ABI 0.64 μm (band 2) level 1B radiances. The full disk view, which covers the entire Earth disk, is a 21696×21696-pixel image given on a fixed grid rectified to the Geodetic Reference System 1980 (GRS80) ellipsoid. The ABI fixed grid is an angle-by-angle coordinate system that represents the vertical near-side perspective projection of the Earth disk from the vantage point of a satellite in an idealized geosynchronous orbit 35,786 km above the Equator (GOES-R PUG L1B Vol 3 Rev 2.2, 2019). The east-west and north-south fixed grid coordinates increase by exactly 14 μrad per pixel in the final resampled image. This 14-μrad instantaneous field of view (IFOV) corresponds to a 500-m ground-projected instantaneous field of view (GIFOV or horizontal spatial resolution) at the equatorial sub-satellite point observed with a view zenith angle (VZA) of ~0º (satellite elevation angle ~90º), as sketched in Fig. 1. At the near-limb locations of Kamchatka and the Kurils, however, the same IFOV corresponds to a GIFOV of ~4 km, because these areas are observed at grazing angles with VZA > 80º (satellite elevation angle < 10º). The ~8 times larger pixel footprint near the limb renders images of horizontal surface features very blurred.

In contrast, the side of a vertically-oriented object, such as a mountain peak or eruption column, is observed at a satellite elevation angle (relative to the side) of almost 90º near the limb—the roles of zenith angle and elevation angle are reversed for vertical orientation. Thus, the local vertically-projected instantaneous field of view (VIFOV or vertical spatial resolution) is only slightly coarser than the equatorial nadir GIFOV, because it scales linearly with distance to the satellite. For example, the Sheveluch volcano in northern Kamchatka is located ~14% further from GOES-17 than the sub-satellite point and observed at VZA = 83.4º, leading to a VIFOV of ~573 m. Thanks to this fine VIFOV, even small vertical features that would be sub-GIFOV were they oriented horizontally can in fact be identified in near-limb images as they protrude from the ellipsoid.

The AHI angular sampling distance in the 0.64 μm visible channel (band 3) is slightly smaller than 14 μrad, resulting in a slightly larger 22000×22000-pixel full disk image (HSD User's Guide v1.3, 2017). The AHI fixed grid is rectified to the World Geodetic System 1984 (WGS84) reference ellipsoid, which is practically identical to the GRS80 ellipsoid for most applications (a tiny difference in flattening leads to a tiny difference in the polar radius). Another small difference is that the ABI full disk image applies a limb mask and excludes space pixels, while the AHI full disk image smoothly transitions into space and does include space pixels; the latter is advantageous for the detection of eruptions very close to the limb.

The ABI data distributed by NOAA have excellent image navigation and registration. The sub-pixel navigation errors are typically 1–2 μrad in both directions, with GOES-17 showing slightly larger errors than GOES-16 (Kalluri et al., 2018; Tan et al., 2019). It should be noted, however, that ABI image navigation performance is at present evaluated mostly for scenes observed with VZA < 75º, in order to filter out low quality measurements resulting from refraction effects and a large GIFOV. Georegistration quality near the limb might therefore be poorer than the quoted values and will need further assessment.




The geolocation accuracy of AHI onboard Himawari-8 is somewhat worse than that of ABI (Takenaka et al., 2020;
Yamamoto et al., 2020). The original AHI data provided by JMA is georegistered based on IR channels with a nominal
resolution of 2 km. There is also a gridded AHI dataset by the Center for Environmental Remote Sensing (CEReS) at Chiba
University, which is more accurately georegistered based on the visible channel with a nominal resolution of 500 m. In this
study, we used CEReS data V20190123, which can have navigation errors of about one band 3 pixel.

Note that a fixed grid image is in a map projection which extremely distorts the shape and area of horizontal features at
the limb. For analyzing vertically-oriented limb objects, however, it is the natural choice as it provides a fairly sharp side
view with minimal foreshortening. Remapping the near-limb portion of a fixed grid image into another projection can give a
considerably distorted view of vertical objects. For example, the common equirectangular projection (also used by NASA
Worldview) introduces severe east-west stretching of higher-latitude mountains and eruption columns (see section 4).

**2.2 Limb volcanoes**

We define an extended limb area in ABI and the very similar AHI full disk images as the band of pixels with VZA > 80°.
This criterion ensures that a near-vertical eruption column protruding from the Earth ellipsoid is observed at close to right
angle. The limb areas for GOES-16, GOES-17, and Himawari-8 are shown in Fig. 2, which also plots the location of
volcanoes that erupted within these regions in the past 100 years. Historic eruption data were obtained from the Holocene
Volcano List of the Global Volcanism Program (2013). The obliquely-viewed volcanic subregions are Iceland for GOES-16,
the Kamchatka Peninsula, the Kuril Islands, the Mariana Islands, Papua New Guinea, southern Chile, and the West Indies for
GOES-17, and the Alaska Peninsula, the southern Indian Ocean, and Antarctica for Himawari-8.

**3. Geometric estimation of eruption column height from one or two satellite views**

Before describing the side view method in detail, we first give a general overview of geometric techniques to retrieve
volcanic eruption column and ash plume height. These techniques will be used in Part 2 of the paper to validate height
estimates by the new method.

As an elementary model of a straight pillar of ash produced by a strong eruption, consider a vertical column of height $h$
with base $B$ and top $T$ protruding from the Earth ellipsoid, locally approximated by the tangent plane, as shown in Fig. 3a.
We define a coordinate system with the origin at $B$, $x$ axis pointing north, and $y$ axis pointing east. The sun-view geometry is
described by the solar zenith and solar azimuth angles $\theta_0$ and $\phi_0$ and the satellite view zenith and view azimuth angles $\theta$ and
$\phi$ (the azimuths correspond to the pixel-to-sun and pixel-to-sensor directions). The sensor-projected location of $T$ in the
satellite image is point $P$, the shadow (or solar projection) of $T$ is cast at point $S$, and the vector from $S$ to $P$ is denoted $\boldsymbol{D}$. The
corresponding view and shadow geometry of a horizontally extended, dispersed plume detached from the surface is
illustrated in Fig. 3b, for simplicity only for the case when the satellite is in the solar principal plane. Here the leading
(farthest from the sun) edge of the plume can be taken as point $T$. The sun-view geometry angles, the sensor- and solar-



projected locations of $T$ and the vector connecting them are known. But base point $B$, that is the nadir projected location of point $T$, is generally unknown. For the special case of a near-vertical column of ejecta, however, the base location can be approximated by the surface $(x,y)$ coordinates of the volcanic vent, which is available from geographical databases.

### 3.1 Method 1: height from sensor-projected length

The above-ellipsoid height of column $\overline{BT}$ can be estimated from its sensor-projected length in the $x$-$y$ plane $\overline{BP}$ and the view
zenith angle $\theta$ simply as

$$h = \frac{\overline{BP}}{\tan \theta}, \tag{1}$$

assuming a flat Earth. Sensitivity to a given error in the measured projected length decreases quickly with increasing view zenith angle. However, the GIFOV also increases rapidly at large $\theta$. For example, a one-pixel error in projected length at $\theta = 84°$ is ~4 km, which translates to a height error of 420 m. In practice, it can be difficult to accurately determine point $P$
at very oblique view angles when using a traditional map projection due to potentially severe image distortions (e.g. east-west stretching in equirectangular projection). For the highest plumes Earth's curvature also has to be accounted for.

### 3.2 Method 2: height from true shadow length

In the same manner as described above, column height can also be estimated from the solar-projected column length (i.e. true shadow) $\overline{BS}$ and the solar zenith angle $\theta_0$ as

$$h = \frac{\overline{BS}}{\tan \theta_0}. \tag{2}$$

This method yields column height above the surface on which the shadow is cast. Eq. (2) corresponds to the simplest case of a flat ocean or flat cloud surface; in the latter case, the absolute height above the ellipsoid can be obtained by adding the estimated cloud height. For shadows over land the calculations are more complex and require a digital elevation model (DEM) to remove topography effects.
180         The height estimate is formally less sensitive to errors in measured shadow length at large solar zenith angles around sunrise or sunset. Determining the end of shadow location, however, can be particularly difficult at these times if the shadow falls near the day-night terminator. Another complication is that at certain sun-view geometries, the length of the *observed* shadow differs from that of the *true* shadow—'true' (stick) shadow length is defined by Eq. (2) as height times the tangent of solar zenith angle. For the special case of a narrow and tall vertical column, the potential difference between the observed and the true shadow lengths can only be negative: when the sun and satellite are on the same side of nadir (small relative
azimuths), part of the true shadow might be obscured by the column itself. For most other sun-view geometries (medium to large relative azimuths), however, the entire true shadow extending from the base of the column is observed. In either case



Eq. (2) can be used, because the starting point of the shadow (the vent location) is known, even when obscured, and, thus, the true shadow length can be determined if the terminus of the shadow is clearly observed.

For a horizontally extended ash layer detached from the surface, on the other hand, the error in the observed shadow length can be both negative and positive, and Eq. (2) is applicable to nadir satellite views only. The case of a suspended ash layer under arbitrary viewing and illumination conditions is discussed in the next section.

### 3.3 Method 3: height from distance between plume edge and shadow edge

The generalization of the stick shadow method to the more common case of a horizontally expanded cloud or ash layer was

derived by Simpson et al. (2000a, b) and Prata and Grant (2001). Here layer height is determined from the direction and length of vector $\boldsymbol{D}$, which connects the terminus of the shadow ($S$) with the sensor-projected image location ($P$) of the leading edge of the plume ($T$). Vector $\boldsymbol{D}$ in this case is the observed (apparent) shadow, whose length is generally different from the true shadow length defined by Eq. (2), as demonstrated in Fig. 3b for the principal plane. For example, if the satellite and the sun are on the same side of the nadir line and $\theta < \theta_0$ (satellite above the sun, sat 1 position) the observed

shadow ($\overline{P_1 S}$) is foreshortened relative to the true shadow ($\overline{BS}$), and if $\theta > \theta_0$ (satellite below the sun, sat 2 position) no leading-edge shadow is observed due to obscuration by the plume. In contrast, if the satellite and the sun are on opposite sides of the nadir line (sat 3 position), the apparent shadow ($\overline{P_3 S}$) is longer than the true shadow, because the satellite also observes shadow cast under the leading edge by other parts of the plume.

    For arbitrary viewing and illumination conditions, points $P$ and $S$ have the following horizontal coordinates on a flat

surface (Fig. 3a):

$$x_P = h \tan \theta \cos(\phi - \pi)$$

$$y_P = h \tan \theta \sin(\phi - \pi)$$

$$x_S = h \tan \theta_0 \cos(\phi_0 - \pi)$$

$$y_S = h \tan \theta_0 \sin(\phi_0 - \pi). \tag{3}$$

Therefore, the components of vector $\boldsymbol{D}$ connecting $S$ to $P$ are

$$x_D = hX$$

$$y_D = hY, \tag{4}$$

where

$$X = \tan \theta \cos \phi - \tan \theta_0 \cos \phi_0$$

$$Y = \tan \theta \sin \phi - \tan \theta_0 \sin \phi_0. \tag{5}$$





The direction (azimuth) and magnitude of $\boldsymbol{D}$ are respectively

$$\phi_D = \arctan\left(\frac{Y}{X}\right) \tag{6}$$

and

$$|\boldsymbol{D}| = h\sqrt{X^2 + Y^2}. \tag{7}$$

Direction $\phi_D$ is independent of $h$, as it is a function solely of the sun-view geometry angles. Once the separation distance $|\boldsymbol{D}|$ between the plume edge and the shadow edge along azimuth $\phi_D$ is determined, the height of the plume edge can be calculated from Eq. (7) as

$$h = \frac{|\boldsymbol{D}|}{\sqrt{X^2 + Y^2}} = \frac{|\boldsymbol{D}|}{\sqrt{\tan^2\theta_0 + \tan^2\theta - 2\tan\theta_0\tan\theta\cos(\phi - \phi_0)}}, \tag{8}$$

where $\phi - \phi_0$ is the relative azimuth angle. Eq. (8) is the generalization of Eq. (2), with the true shadow length $(\overline{BS})$ in the
numerator being replaced by the apparent shadow length and $\tan\theta_0$ in the denominator being replaced by a more complicated formula, which depends on the view zenith and relative azimuth angles too.

Note that for a horizontal suspended ash layer only a single height estimate can be derived from the (edge) shadow using Eq. (8), which becomes Eq. (2) for nadir viewing. A vertical column, however, represents a special case, for which two separate height estimates can be derived from the shadow, using both Eq. (2) and Eq. (8). This is so because for a column,
the surface projected location of top point $T$ is known from three different directions: the satellite view (point $P$), the solar view (point $S$, the shadow terminus), and the effective nadir view (base point $B$). The vector (parallax) between any two of these surface projections can be used, in conjunction with the sun-view angles, to estimate the height: method 1 for $\overline{BP}$ (sensor projected length), method 2 for $\overline{BS}$ (true shadow length), and method 3 for $\overline{PS}$ (the 'apparent shadow', although this line segment is not in shadow for a column). For a suspended ash layer, however, the nadir-projected base point $B$ is
unknown and thus only method 3 is applicable.

In the practical implementation of Eq. (8), the satellite image is first rotated so that one of its axes aligns with azimuth $\phi_D$ and then the plume–shadow separation distance can be easily calculated along an image row or column. The plume and shadow edges can be delineated visually by a human observer or more objectively by a wide variety of edge detection algorithms (Canny, Roberts, Sobel, Prewitt, Laplacian, etc.). As before, a DEM is needed to remove topography effects
when shadows over land are analyzed.

### 3.4 Method 4: height from stereoscopy

The previous three methods estimate plume height from a single satellite view. This is possible in two special cases when an extra piece of information can be recovered from the image in addition to the sensor-projected plume top location. One, for vertical eruption columns the location of the base (i.e. the nadir-projected location of the top) can be approximated by that of



the volcanic vent. Two, when shadows are visible they provide the projected location of a plume top/edge point from a second (the solar) perspective. Therefore, these single-image methods can be considered as effective 'stereo' methods, because they use the surface locations of a plume point projected from two different view directions.

In the general case, column/plume height can be estimated by proper stereoscopy utilizing views from at least two different satellites. Note that the formulas derived in section 3.3 can also be used as a simplified stereo algorithm, if the

shadow terminus location and the solar zenith/azimuth angles are replaced respectively by the sensor-projected plume location and the view zenith/azimuth angles corresponding to a second satellite. In this case $D$ is the parallax vector between the two satellite projections. Applying the generalized shadow method in stereo mode implies the assumptions that (i) the satellite images are perfectly time synchronized and (ii) the two look-vectors, which connect the projected plume locations to the corresponding satellites, have an exact intersection point. If the images are asynchronous, plume advection between the

acquisition times has to be corrected for. Furthermore, look-vectors never intersect in practice due to pixel discretization and image navigation uncertainties. Therefore, dedicated stereo algorithms use vector algebra to search for the height that minimizes the distance between the passing look-vectors rather than rely on the analytical solution derived from view zenith and azimuth angles with the assumption of exact line intersection.

In Part 2, we use both the shadow-stereo method and a dedicated stereo code for validation. The analytical solution of

method 3 is applied to plume top features that could be visually identified in both GOES-17 and Himawari-8 images. Limb imagery is generally unsuitable for automated stereo calculations due to the difficulty of pattern matching between an extreme side view and a less oblique view. A human observer, however, can still identify the same plume top feature even in such widely different views.

We also use a novel fully-automated stereo code to retrieve plume heights of the 2019 Raikoke eruption. This "3D

Winds" algorithm, which was originally developed for meteorological clouds, combines geostationary imagery from GOES-16, GOES-17, or Himawari-8 with polar-orbiter imagery from the Multiangle Imaging SpectroRadiometer (MISR) or Moderate Resolution Imaging Spectroradiometer (MODIS) to derive not only plume height but also the horizontal plume advection vector (Carr et al., 2018, 2019; Horváth et al., 2020). The technique requires a triplet of consecutive geostationary full disk images and a single MISR or MODIS granule. Feature templates are taken from the central repetition of the

geostationary triplet and matched to the other two repetitions 10 min before and after, providing the primary source of plume velocity information. The geostationary feature template is then matched to the MISR or MODIS granule which is observed from a different perspective, providing the stereoscopic height information. The apparent shift in the pattern from each match, modeled pixel times, and satellite ephemerides feed the retrieval model to enable the simultaneous solution for a horizontal advection vector and its geometric height.

**3.5 Note on potential azimuth distortions in mapped satellite images**

At this point, it is worthwhile to note the possibility of angular distortions in a map projected satellite image, because this caveat is usually ignored in geometric height retrievals. Methods 1 and 2 require respectively the sensor-projected column





length along the view azimuth $\phi$ and the shadow length along the solar azimuth $\phi_0$, while method 3 requires the distance
between the plume edge and the shadow edge along the azimuth $\phi_D$. If the image is in a non-conformal map projection,
these angles are not preserved locally.

    The equirectangular projection (Plate-Carrée), used in NASA Worldview and also implicit in the gridded CEReS AHI
data, is a non-conformal projection that has a constant meridional scale factor of 1 but a parallel (zonal) scale factor that
increases with latitude $\varphi$ as $\sec(\varphi)$. The non-isotropic scale factor leads to considerable east-west stretching and azimuth
distortion at the latitudes of Kamchatka and the Kurils. The magnitude of angular distortion depends on azimuth and can
easily be 10°. Angular distortion has to be considered or a locally conformal map projection needs to be used when applying
methods 1–3.

## 4 Side view method

### 4.1 Measurement principle

This method is essentially the same as method 1, but instead of calculating linear distances in a conventional map projection,
it determines the angular extent of an eruption column from the ABI fixed grid image. Operating in the angular space of the
fixed grid has the advantages of (i) working with a more 'natural', less distorted view of a protruding column (e.g. no zonal
stretching), (ii) the VIFOV being considerably smaller than the GIFOV, and (iii) no Earth curvature effects. The
geostationary side view geometry of the measurement is sketched in Fig. 4. In the following we ignore atmospheric
refraction effects, which will be shown to be largely negligible in section 4.2.

The satellite coordinate system has its origin located at the satellite's center of mass. The $x$ axis ($S_x$) points from the
satellite to the center of the Earth and the upward pointing $z$ axis ($S_z$) is parallel to the line connecting the center of the Earth
with the North Pole. The $y$ axis ($S_y$) is aligned with the equatorial axis and completes the right-handed coordinate system.

    The look vectors connecting the satellite to base $B$ and to the ellipsoid projected location of top $T$, that is $P$, are
respectively $\boldsymbol{S_B} = (S_{B,x}, S_{B,y}, S_{B,z})$ and $\boldsymbol{S_P} = (S_{P,x}, S_{P,y}, S_{P,z})$. For convenience, Fig. 4 depicts an eruption column on the
meridian of the sub-satellite point and thus plots the ellipsoidal cross section along the $S_x$–$S_z$ plane. In the general case, the
shown cross section corresponds to the cutting plane defined by the $S_x$ axis and look vector $\boldsymbol{S_B}$, which is obtained by rotating
the $S_x$–$S_z$ plane around the $S_x$ axis by the geocentric colatitude. The image location of $B$ is determined from the known
geodetic latitude and longitude of the volcano. Calculation of the satellite-to-pixel look vector and the geodetic latitude and
longitude of a given ABI image pixel is described in Appendix A.
The angle $\delta$ between the top and base look vectors can be determined from the dot product of $\boldsymbol{S_B}$ and $\boldsymbol{S_P}$ as

$$\delta = \mathrm{acos}\left(\frac{\boldsymbol{S_B} \cdot \boldsymbol{S_P}}{|\boldsymbol{S_B}||\boldsymbol{S_P}|}\right). \tag{9}$$

If $\delta$ is expressed in radians, an estimate of column height $h$ can be obtained simply as





$$\hat{h} = \delta |S_B|. \tag{10}$$

This estimate is the projection of the column height to the axis $\hat{Z}$, which is perpendicular to look vector $S_B$ and tilted slightly

from the local vertical axis $Z$ by an angle of $90° - \theta$. The projection of $T$ onto the $\hat{Z} - S_B \times \hat{Z}$ plane by look vector $S_P$ is

point $\tilde{T}$. The foreshortening due to the slight rotation from the exact limb ($\theta = 90°$) can be corrected by dividing $\hat{h}$ by

$\cos(90° - \theta)$, which is an almost trivial correction of 154 m / 10 km at $\theta = 80°$ and 55 m / 10 km at $\theta = 84°$. Note that the

angular distance between the geodetic zenith and the geocentric zenith (or the angle of the vertical) equals the difference

between the geodetic latitude and the geocentric latitude, which is ~0.18° for Kamchatka. This is a negligible difference for

our purposes and thus axis $Z$ can be either the geodetic or the geocentric vertical.

The horizontal expansion of the plume top in the radial direction, or equivalently the radial tilt of an eruption column

with no umbrella cloud, can however introduce substantial biases. Expansion of the umbrella cloud towards the limb (away

from the satellite), depicted by red in Fig. 4, leads to an overestimated angle $\delta_+$ between the base and top look vectors and

positive height bias. Conversely, plume expansion away from the limb (towards the satellite), depicted by blue in Fig. 4,

leads to an underestimated $\delta_-$ and negative height bias. To minimize such biases for an expanded umbrella cloud, one has to

estimate the plume point closest to the vertical at the vent and use that as point $T$ in the calculations. If under weak winds the

plume expansion is fairly isotropic and advection is small, the center of the plume—determined visually or by fitting a circle

to the umbrella cloud—is a good choice for $T$ (see the top middle inset in Fig. 4). Under strong winds, a point near the

windward plume edge might be used instead. The approximate nature of locating the plume point with the smallest radial

distance to the vent, however, introduces an inevitable uncertainty in the height estimate.

In contrast, the tilt of the eruption column relative to $\hat{Z}$ in the $\hat{Z} - S_B \times \hat{Z}$ plane can be corrected for (see the top right

inset in Fig. 4). The sideways tilt angle $\beta$ can be determined from the dot product of $\hat{Z}$ and vector $V_{B\tilde{T}}$ connecting base $B$ to

$\tilde{T}$:

$$\beta = \text{acos}\left(\frac{\hat{Z} \cdot V_{B\tilde{T}}}{|\hat{Z}||V_{B\tilde{T}}|}\right), \tag{11}$$

where

$$\hat{Z} = S_B - \left(h_{sat} + r_{eq} - |V_{CB}| \tan(90° - \theta), 0, 0\right),$$

$$V_{CB} = S_B - \left(h_{sat} + r_{eq}, 0, 0\right),$$

$$V_{B\tilde{T}} = S_P \frac{|S_B|}{|S_P|} - S_B. \tag{12}$$

Here $h_{sat} = 35,786,023$ m is the satellite (perspective point) height above the ellipsoid, $r_{eq} = 6,378,137$ m is the semi-

major axis of the GRS80 ellipsoid, and vector $V_{CB}$ connects the center of the ellipsoid to base $B$. The final height estimate is

then obtained as



$$\hat{h}_f = \hat{h} \frac{\cos(\beta)}{\cos(90° - \theta)}. \tag{13}$$

In the actual implementation of the algorithm, we up-sample ABI images by a sub-pixel factor (SPF) of two using bilinear interpolation. This follows the exact practice of the operational ABI image navigation and registration assessment
tool, which also refines ABI images to half a pixel resolution (Tan et al., 2019). Here it is relevant to note that the native spatial sampling by the ABI detectors (10.5–12.4 µrad) is finer than the pixel resolution (14 µrad) of the level 1B product (Kalluri et al., 2018). Strictly for visual clarity, images in subsequent figures are magnified further; however, all height calculations are performed on data up-sampled with SPF = 2.

## 4.2 Refraction effects

The geometry of terrestrial refraction is sketched in Fig. 5. For spaceborne observations, the known quantity is the unrefracted view zenith angle of a pixel, which is the zenith angle at the intersection of the idealized (prolonged exoatmospheric) ray with the Earth ellipsoid (point $P$). An actual outgoing ray ascending through the atmosphere is refracted away from the zenith; hence, its zenith angle increases with height as it slightly bends toward the satellite sensor. Consequently, the apparent image position of the terrestrial source of the ray is displaced in the radial direction to a point
with a larger (unrefracted) view zenith angle, that is, closer to the limb (i.e. from point $P'$ to point $P$). A grazing ray traverses a substantial range in latitude and/or longitude and is subject to fluctuations in weather, which are complicated to handle properly. Most Earth remote sensing applications, however, can rely on the analytical treatment of refraction by Noerdlinger (1999), which was designed to derive corrections to geolocation algorithms.

This method assumes a spherical Earth and spherically symmetric atmosphere and relates three different angles by simple
analytical formulas: the known zenith angle $\theta$ of the unrefracted ray at surface point $P$, the zenith angle $\hat{\theta}$ of the same unrefracted ray at the vertical of the true (refracted) Earth intersection point $P'$, and the zenith angle $\theta'$ of the refracted ray at surface point $P'$. (In the notation of Noerdlinger (1999) these angles are respectively $z_0$, $z$, and $z'$.) In general, $\theta > \hat{\theta} > \theta'$. For the calculation of the bidirectional reflectance distribution function, the zenith angle difference $\theta - \theta'$ is the relevant refraction effect. For the calculation of the apparent horizontal displacement of the observed point, i.e. the distance between
$P'$ and $P$, however, the zenith angle difference $\theta - \hat{\theta}$ is needed.

The calculations by Noerdlinger (1999) for 'white light' (0.46–0.53 µm), sea level, a surface temperature of 15°C, and a refractive index at the surface of $\mu = 1.000290$ yield a linear displacement of 448 m at $\theta = 80°$ and a range of 1267–4858 m at $\theta = 83°$–86° typical for Kamchatka and the Kurils; these numbers could vary by ~25% depending on weather. Using the Ciddor (1996) equation for the refractive index—which is the current standard and is a function of wavelength,
temperature, pressure, humidity, and $CO_2$ content—slightly reduces the surface refractive index at 0.64 µm to $\mu = 1.000277$ and the original Noerdlinger (1999) displacement values by a few dozen to a few hundred meters. Considering that the band



2 GIFOV (or ground sample distance) rapidly increases near the limb and reaches ~4 km at $\theta = 84°$, the horizontal displacement due to refraction could be a relatively modest 1–2-pixel shift at the surface.

More importantly, however, the horizontal displacement of the surface base point does not affect our height estimates, as long as ABI pointing and angular sampling are stable, because the pixel location of $B$ is fixed by the known geodetic latitude/longitude of the volcano, rather than by searching for the (potentially shifted) position of the volcano's feature template within the full disk image. The height calculation is only affected by a shift in the position of top $T$, because $T$ has to be located in the image by visual means. The refractive displacement scales as $(\mu - 1.0)$, which in turn is approximately proportional to pressure. Therefore, the linear displacement at 5 km altitude (~500 hPa) is about half of the surface value and less than a third of that at the tropopause. This amounts to a practically negligible sub-pixel shift for plume tops above 5 km. For plume tops below 5 km, one might apply a general 1-pixel radial shift away from the limb; such a first-order correction works well for the Kamchatka volcanic peaks we use in the next section to validate the height retrieval technique.

We note that refraction is not considered in the operational ABI image navigation, because its effect has been deemed marginal compared to the variation of measured geolocation errors (Tan et al., 2019). We also note for completeness that refraction changes the apparent solar zenith angle at low Sun and thus also affects the shadow-based height estimation methods through $\tan\theta_0$. In this case, the relevant quantity is the angle difference $\theta_0 - \theta_0'$, which is ~0.19° at $\theta_0 = 84°$, causing a relatively small ~300 m / 10 km underestimation.

**4.3 Validation using volcanic peaks**

Volcanic peaks in the Kurils and Kamchatka provide a large set of static targets for the validation of the side view height estimation method. Figure 6 exemplifies the GOES-17 fixed grid view of central Kamchatka on two different days, when several volcanic peaks were clearly visible. The images demonstrate that viewing conditions, what astronomers call 'seeing', can vary significantly from day to day and even diurnally, because of changing turbulence, lighting, and haziness. Therefore, a given mountain peak can easily be identified on certain days but not on others.

Figure 7 shows magnified views of three volcanoes of increasing summit elevation: Alaid (2339 m), Kronotsky (3528 m), and Kamen (4619 m). Here the base point (red diamond) was fixed by the geodetic coordinates of the vent, while the peak position (blue diamond) was visually determined and then corrected for refraction by applying a 1-pixel inward radial shift; that is, the top pixel $T$ used in the height calculations is the one located 1 pixel 'below' the visual peak in the rotated images. Note that the distance between the base and peak pixel increases with increasing summit elevation. Figure 7c also highlights that identification of peaks is often the easiest when the volcanoes peek through a lower level cloud layer.

Figure 8 demonstrates the height estimation through the example of Kronotsky. Using the marked base and top pixels, the side view method yields a height estimate of 3548 m, which is in excellent agreement with the true height of 3528 m. Figure 8a shows the fixed grid view with base-relative isoheight lines drawn as visual aid. These contour lines were obtained by calculating the height between the base pixel and every single image pixel as described in section 4.1. In practice, this amounts to determining the intersection of a given pixel's look vector and the plane that contains the base point and is





perpendicular to the base pixel's look vector (i.e. the $\widehat{Z} - S_B \times \widehat{Z}$ plane in Fig. 4). The visually identified top pixel is located correctly between the 3 km and 4 km contour lines.

The GOES-17 image in the traditional equirectangular projection is plotted in Fig. 8b. The mapped image is severely distorted compared to the natural fixed grid view, as it is stretched in the parallel (east-west) direction by a factor of $\sec(\varphi = 54.75°) = 1.73$. This makes image interpretation and the precise identification of the peak more difficult. The non-

isotropic scale factor also leads to considerable azimuth distortions. Although the true GOES-17 view azimuth is $\phi = 113°$, the mapped image of the volcano lies along the apparent (distorted) azimuth of $\phi = 103°$. The ellipsoid projected distance between the base and our best visual estimate peak location is ~31 km, which yields a height of 3768 m using Eq. (1) with a view zenith angle of $\theta = 83.07°$. Although the height estimate is fairly decent in this particular case, the severe image distortions make height calculation from linear distances measured in a conventional map projection generally inferior to the

angular technique facilitated by the fixed grid side view.

For a more comprehensive validation, we selected 50 mountain peaks ranging in elevation from 502 m (Mashkovtsev) to 4835 m (Klyuchevskoy). The geodetic latitude/longitude and the elevation of a given peak can differ slightly between databases (Global Volcanism Program, Google Earth, KVERT list of active volcanoes), due mainly to the use of different reference ellipsoids (the exact choice of which, however, is usually undocumented). For example, the elevation of Kronotsky

is 3482 m in the Global Volcanism Program database and 3528 m in the KVERT list. In our work, we relied on PeakVisor (https://peakvisor.com), which is one of the most advanced mountain identification and 3D maps tool (also available as a mobile app). Its peaks database has almost a million named summits and is based on DEMs from the European Copernicus program, the United States Geological Survey (USGS), and the Japan Aerospace Exploration Agency (JAXA).

The name, geodetic latitude and longitude, true height, and estimated height of the selected peaks, as well as the date and

time of the GOES-17 images used are listed in supplementary Table S1. As previously discussed, the observing conditions show considerable temporal variation, but for a static target one has the luxury of a large number of available images from which to choose one that offers a good view of the peak. For a volcanic plume, one is limited to a few images around the eruption time; however, identifying a high-altitude plume top is also much easier, because reduced atmospheric turbulence and the distinct color of the ejecta usually lead to good contrast against the background. The height error estimates obtained

below for volcanic peaks under good 'seeing', nevertheless, represent a lower limit, because the (typically unknown) radial tilt of eruption columns introduces additional uncertainty in the retrievals.

The scatter plot of estimated peak height versus true peak height is given in Fig. 9 (remember, ABI images are up-sampled with SPF = 2). For each data point, we performed a sensitivity analysis by shifting the visually determined peak location by ±1 pixel in either direction and calculating heights for the central pixel and its 8-pixel neighborhood. The

standard deviation of these nine height values is then used as error bar. The root mean square error (RMSE) computed for the 50 peaks corresponds to the unperturbed best estimate peak locations. As shown, the overall bias is a negligible 28 m and the RMSE is 150 m. The error bar on individual retrievals is ~250 m, with a maximum possible height discrepancy of ±400 m



due to a 1-pixel uncertainty in peak location; these numbers increase to ~300 m and ±500 m when ABI images are used without up-sampling (SPF = 1). We conclude from these results that in the absence of significant radial tilt, ±500 m is a
reasonable uncertainty value for instantaneous height retrievals.

## 4.4 Sheveluch eruption on 8 April 2020

As a final example in Part 1, we demonstrate the side view method on the 8 April 2020 eruption of Sheveluch. A detailed analysis of this case, including a comparison with other height retrieval methods, is given in Part 2. The Sheveluch volcano has three main elements: Old Sheveluch with an elevation of 3307 m, the old caldera, and the active Young Sheveluch with a
lava dome at 2589 m surrounded by peaks of about 2800 m elevation. A strong explosive eruption occurred on 8 April 2020 at 19:10 UTC, slightly after sunrise, whose ash plume advected south-southeast from Young Sheveluch. The Kamchatka Volcanic Eruption Response Team issued an orange coded Volcano Observatory Notice for Aviation (VONA 2020-40, http://www.kscnet.ru/ivs/kvert/van/?n=2020-40), reporting a plume height of 9.5–10.0 km as determined by the basic satellite temperature method from Himawari-8 11 μm data.
The GOES-17 fixed grid images capturing the eruption are presented in Figure 10. Figure 10a shows the volcano at 19:00 UTC before the eruption, with the volcanic peaks, especially the more northerly Old Sheveluch, clearly recognizable above a low-level stratus layer. Far away peaks of the Sredinny Mountain Range west of Sheveluch can also be seen in the background. By the time of the next full disk image acquired at 19:10 UTC and depicted in Fig. 10b, the eruption column reached its maximum altitude and developed a small umbrella cloud, which was advected slightly to the south-southeast by
the prevailing upper-level winds. Notable features in this image are a portion of the long shadow cast by the ash column and the brighter sunlit near side (eastern edge) and the darker shadowed far side (western edge) of the umbrella cloud, brought out by the rising Sun in the east ($\theta_0 = 86.0°, \phi_0 = 82.6°$).

A simple technique commonly applied to aid change detection in multitemporal imagery is the computation of running-difference images. The normalized running-difference (RD) image at time $t$ can be defined as

$$image_{RD,t} = image_t - image_{t-1} \frac{\langle image_t \rangle}{\langle image_{t-1} \rangle}, \tag{14}$$

where *image* is a 2D array of reflectances and $\langle \rangle$ indicates the mean of all pixels. The advantage of such a running-difference image, whose mean pixel value is ~0, is that static or quasi-static background features are removed, making identification of dynamic features easier. The RD image calculated from the 19:10 UTC and 19:00 UTC GOES-17 snapshots is plotted in Fig. 10c. Note how the plume and its shadow are accentuated, while Old Sheveluch and the peaks of the Sredinny mountains
are removed in this image. Similar or more sophisticated image change detection algorithms (Radke et al., 2005) could later be used for the automated detection of volcanic eruptions in side view imagery, where traditional methods based on brightness temperature differences might be problematic due to the extreme view geometry (e.g. limb darkening or brightening effects).





Finally, Fig. 10d presents the fixed grid view of the plume at 19:10 UTC with the base-relative isoheight lines drawn.
The summit of Old Sheveluch is correctly located between the 3 km and 4 km contour lines. Our best estimate plume top position directly above the vent is marked by the blue diamond, which was visually determined considering the expansion and slight advection of the umbrella cloud and which lies halfway between the far-side and near-side edges. The selected top pixel leads to a plume height estimate of ~8 km. Note that the radial expansion of the plume at an approximately constant altitude results in height biases as sketched in Fig. 4. The darker far-side edge of the umbrella, which is located behind the
plane of the isoheights, appears at an altitude of ~9 km, while the brighter near-side edge, which is located in front of the isoheights' plane, appears at ~7 km. In contrast, the latitudinal (left-right) expansion of the plume has little effect on the height estimates. The side view plume height of ~8 km is 1.5–2.0 km lower than the height from the temperature method and, based on the discussion above, we think it is closer to the true plume altitude. This discrepancy, caused by well-known retrieval biases near the tropopause temperature inversion, is further investigated in Part 2.

**5 Summary**

We presented a simple geometric technique that exploits the generally unused near-limb portion of geostationary fixed grid images to estimate the height of a volcanic eruption column. Such oblique angle observations provide an almost orthogonal side view of a vertical column protruding from the Earth ellipsoid and allow height calculation by measuring the angular extent of the column. We demonstrated the technique using data from the ABI instrument, which offers the highest
resolution visible imagery and most accurate georegistration among current generation geostationary imagers. The publicly available ABI level 1B data distributed by NOAA also contain all the information required for the calculations. The main volcanic regions that reside close to the limb of the ABI and the very similar AHI instruments and which are thus amenable to the side view analysis are Iceland (GOES-16), Kamchatka, the Kuril Islands, Papua New Guinea, southern Chile (GOES-17), and Alaska (Himawari-8).

Thanks to its purely geometric nature, the technique avoids the major pitfalls of the traditional brightness temperature method (thermal disequilibrium, semitransparency effects, uncertainty in ash chemical composition, sensitivity to the assumed temperature profile and ambiguity at the tropopause temperature inversion); however, it is mainly applicable to strong eruptions with nearly vertical columns and its coverage is limited to daytime point estimates in the immediate vicinity of the vent. Nevertheless, the side view technique can provide complementary constraints on source conditions for the
modeling of near-field plume behavior. The longer slant path of the oblique view also increases the reflected signal, which can lead to enhanced detection of optically thin plumes that are semi-transparent from an overhead perspective.

Initial validation of the technique on mountain peaks in Kamchatka and the Kurils indicates that ±500 m is a reasonable preliminary uncertainty value for height estimates of near-vertical eruption columns, although the practically achievable uncertainty will have to be determined by further studies of actual plumes. This height uncertainty compares favorably with
the 2–4 km uncertainty typical of state-of-the-art radiometric methods. The radial expansion of the volcanic umbrella cloud



or the radial tilt of a weak eruption column under strong wind shear, however, introduces additional errors that need further characterization. To minimize such retrieval errors, in the current proof-of-concept study we visually estimated the plume top point that is closest to the local vertical at the vent. The technique can later be improved by using reanalysis or forecast wind profiles to estimate the radial spreading of the umbrella cloud or the radial tilt of heavily bent eruption columns and
rotate the plane of the drawn isoheights into the prevailing wind direction.

**Appendix A**

**A1 Calculation of ABI look vectors and geodetic coordinates**

The formulas to compute the geodetic coordinates and the look vector of an ABI fixed grid image pixel are provided here for convenience. The description follows section 5.1.2.8 of the GOES R Series Product Definition and Users' Guide (GOES-R
PUG L1B Vol 3 Rev 2.2, 2019), which is the definitive reference. The corresponding formulas for AHI can be found in section 4.4 of the Low Rate Image Transmission (LRIT) / High Rate Image Transmission (HRIT) Global Specification (CGMS, 2013).

The coordinate systems used for image navigation are illustrated in Fig. A1. The Earth Centered Fixed (ECF) coordinate system rotates with the Earth and has its origin at the center of the Earth. Its $x$ axis ($e_x$) passes through the intersection of the
Greenwich Meridian and the Equator, while its $z$ axis ($e_z$) passes through the North Pole. Its $y$ axis ($e_y$) is defined as the cross product of the $z$ axis and the $x$ axis to complete the right-handed coordinate system.

The satellite coordinate system has its origin at the satellite's center of mass. Its $x$ axis ($S_x$) points from the satellite to the center of the Earth and its $z$ axis ($S_z$) is parallel to the ECF $z$ axis ($e_z$). Its $y$ axis ($S_y$) is aligned with the equatorial axis and completes the right-handed coordinate system.
For point $P$ on the GRS80 ellipsoid, the geocentric latitude $\varphi_c$ is the angle between the local radius vector and the equatorial plane, while the geodetic latitude $\varphi$ is the angle between the local surface normal and the equatorial plane. The geodetic latitude, which is the one used in image navigation, is larger than the geocentric latitude at all locations except the poles and the Equator where they are equal (the maximum difference between $\varphi$ and $\varphi_c$ is ~0.19°). The geodetic and geocentric longitudes $\lambda$ are however the same. Longitude is positive east of and negative west of the Greenwich Meridian.
The notation for the scan angles that describe the position of a pixel within the fixed grid is somewhat confusing though. In the L1B netCDF data files, the east-west (horizontal) scan variable is called '$x$' and the north-south (vertical) scan variable is called '$y$', both given in radians. Scan angle '$x$' is negative west of and positive east of the subsatellite longitude, and scan angle '$y$' is positive north of and negative south of the Equator. The north-south ('$y$') angle elevates the east-west ('$x$') scan plane relative to the equatorial plane.
With these caveats in mind, the satellite-to-pixel look vector $\boldsymbol{S_P} = \left(S_{P,x}, S_{P,y}, S_{P,z}\right)$ is given in the satellite coordinate system as





$$S_{P,x} = r_s \cos(x) \cos(y)$$

$$S_{P,y} = -r_s \sin(x)$$

$$S_{P,z} = r_s \cos(x) \sin(y), \tag{A1}$$

where

$$r_s = |S_P| = \frac{-b - \sqrt{b^2 - 4ac}}{2a} \tag{A2}$$

is the distance between the satellite and point $P$, and

$$a = \sin^2(x) + \cos^2(x)\left[\cos^2(y) + \frac{r_{eq}^2}{r_{pol}^2}\sin^2(y)\right]$$

$$b = -2H \cos(x) \cos(y)$$

$$c = H^2 - r_{eq}^2. \tag{A3}$$

Here $r_{eq} = 6{,}378{,}137$ m and $r_{pol} = 6{,}356{,}752.31414$ m are the semi-major and semi-minor axes of the GRS80 ellipsoid, $h_{sat} = 35{,}786{,}023$ m is the satellite height above the ellipsoid, and $H = h_{sat} + r_{eq} = 42{,}164{,}160$ m is the satellite distance from the center of the Earth; all of these parameters are included in the L1B product files.

The geodetic latitude and longitude of point $P$ are then computed as

$$\varphi = \arctan\left(\frac{r_{eq}^2}{r_{pol}^2} \frac{S_{P,z}}{\sqrt{(H - S_{P,x})^2 + S_{P,y}^2}}\right)$$

$$\lambda = \lambda_0 - \arctan\left(\frac{S_{P,y}}{H - S_{P,x}}\right), \tag{A4}$$

where the longitude of the nominal subsatellite point is $\lambda_0 = -75°$ for GOES-16 and $\lambda_0 = -137°$ for GOES-17; a negative number because both satellites are positioned west of the Greenwich Meridian.

*Data availability.* The GOES-R ABI L1B radiances are available from the NOAA Comprehensive Large Array-data
Stewardship System (CLASS) archive (https://www.avl.class.noaa.gov/saa/products/welcome). Himawari 8/9 gridded data are distributed by the Center for Environmental Remote Sensing (CEReS), Chiba University, Japan (http://www.cr.chiba-u.jp/databases/GEO/H8_9/FD/index.html). The Holocene Volcano List is compiled by the Global Volcanism Program: Volcanoes of the World, v. 4.9.1. Venzke, E (ed.), Smithsonian Institution, https://doi.org/10.5479/si.GVP.VOTW4-2013. The KVERT Volcano list is available at http://www.kscnet.ru/ivs/kvert/volcano.php?lang=en. VolSatView uses the



following dataset: National Centers for Environmental Prediction/National Weather Service/NOAA/U.S. Department of Commerce, European Centre for Medium-Range Weather Forecasts, and Unidata/University Corporation for Atmospheric Research: Historical Unidata Internet Data Distribution (IDD) Gridded Model Data, Research Data Archive at the National Center for Atmospheric Research, Computational and Information Systems Laboratory, https://doi.org/10.5065/549X-KE89, 2003. The PeakVisor summit database (freely searchable on the website) and mobile app (requiring a subscription) are
available at https://peakvisor.com.

*Author contributions*. ÁH developed the idea and methodology of the side view retrievals during discussions with GAH and SAB. Retrievals from the 3D Winds stereo code were provided by its developers JLC and DLW, while retrievals from the VolSatView information system were provided by its creators AAB, AAM, OAG, and DVM. ÁH analyzed the results and prepared the manuscript with significant contributions from all authors.

*Competing interests*. The authors declare that they have no conflict of interest.

*Special issue statement*. This article is part of the special issue "Satellite observations, in situ measurements and model simulations of the 2019 Raikoke eruption (ACP/AMT/GMD inter-journal SI)". It is not associated with a conference.

*Acknowledgements*. ÁH, GAH, and SAB are members of the VolPlume project within the research unit VolImpact funded by the German Research Foundation DFG (FOR 2820). This work also contributes to the Cluster of Excellence "CLICCS—
Climate, Climatic Change, and Society" funded by the Deutsche Forschungsgemeinschaft DFG (EXC 2037, Project Number 390683824), and to the Center for Earth System Research and Sustainability (CEN) of Universität Hamburg.

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



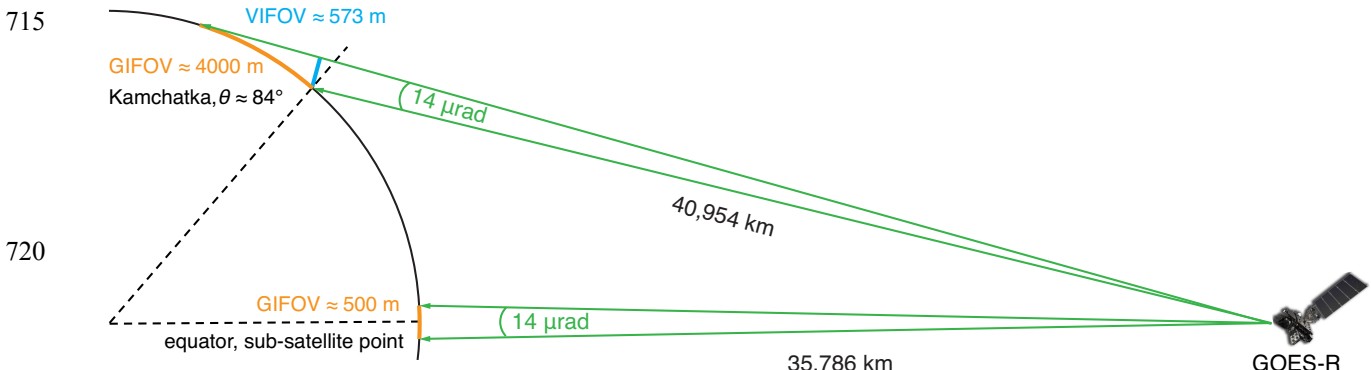

**Figure 1.** Horizontal (GIFOV, orange) and vertical (VIFOV, blue) spatial resolution of an ABI band 2 fixed grid pixel at the sub-satellite point and the Sheveluch volcano in Kamchatka observed at a view zenith angle of $\theta \approx 84°$. The fixed grid has an angular resolution of 14 µrad in both the east-west and the north-south directions and is rectified to the GRS80 ellipsoid as viewed from an idealized geostationary position. Note the figure is not drawn to scale.



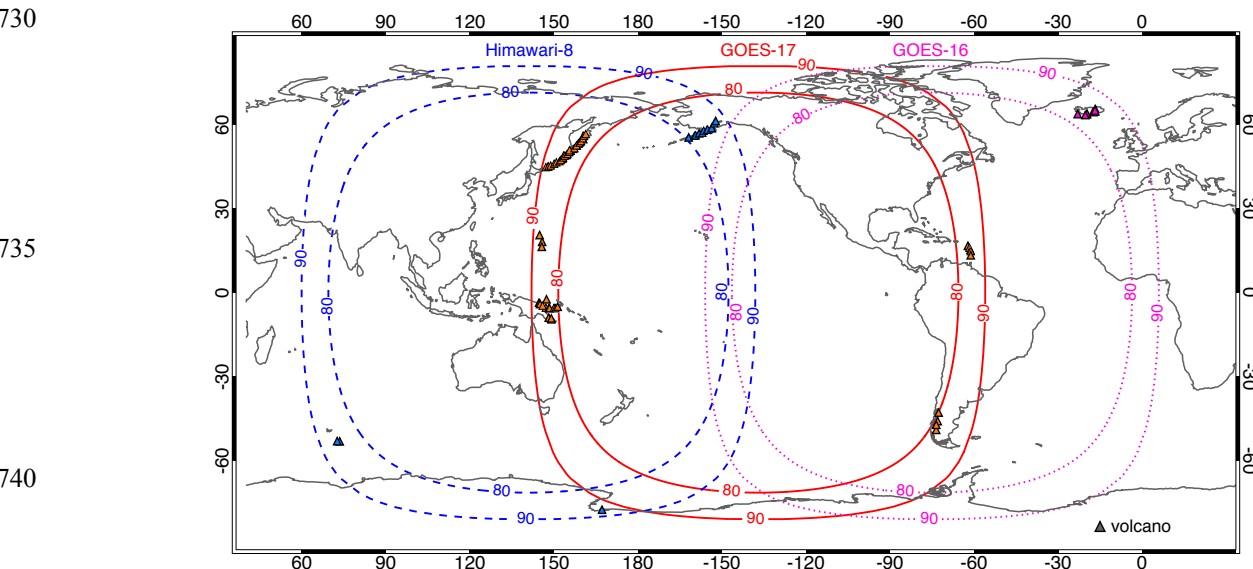

**Figure 2.** Limb area between the 80º and 90º view zenith angle isolines of the GOES-16 (dotted magenta), GOES-17 (solid red), and Himawari-8 (dashed blue) full disk. Triangles indicate volcanoes that erupted within the limb areas in the past 100 years.





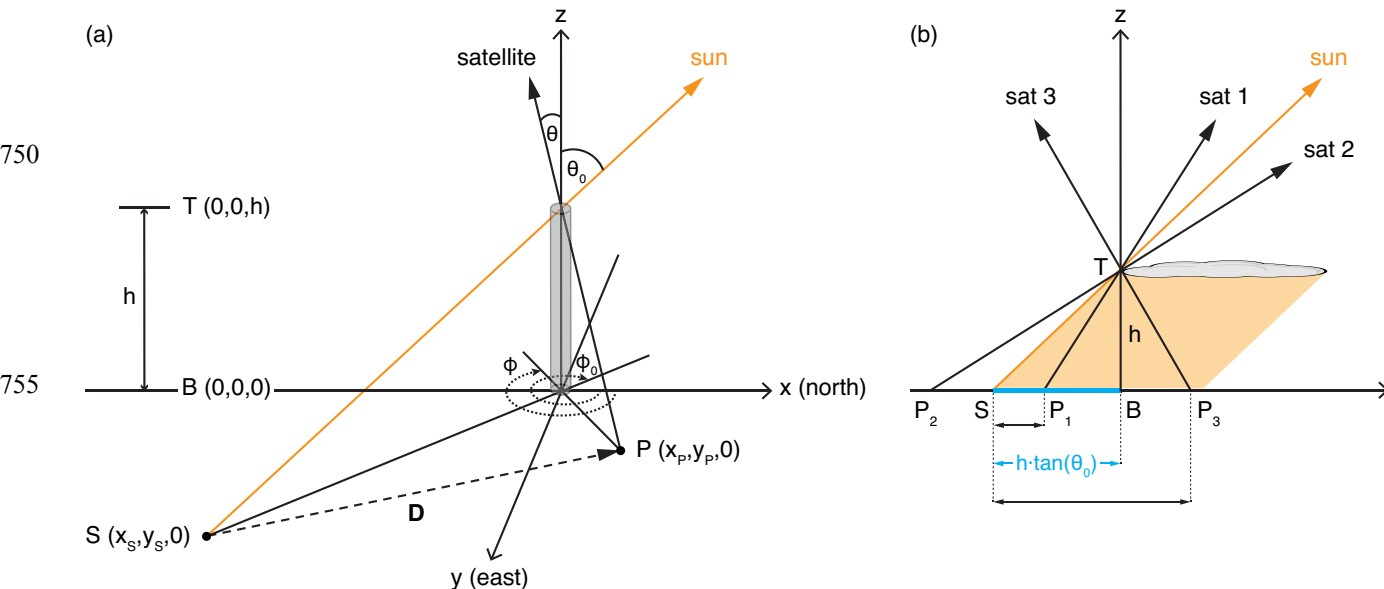

**Figure 3.** Sun-satellite-shadow geometry for **(a)** a narrow vertical eruption column under arbitrary viewing and illumination conditions and **(b)** a horizontally expanded suspended ash layer in the solar principal plane of panel **(a)**. In panel **(b)**, the cyan line segment represents the true (stick) shadow length and the orange coloring indicates the area in the shadow of the ash layer.





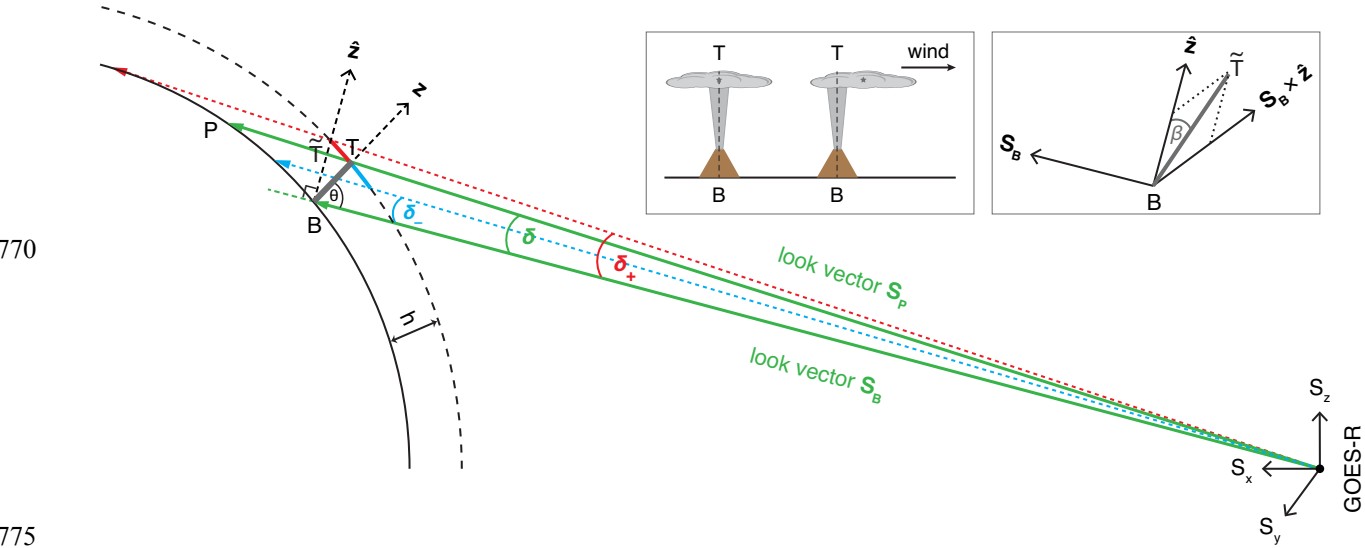

**Figure 4.** Side view geometry of a vertical column located near the limb as imaged by a geostationary sensor.






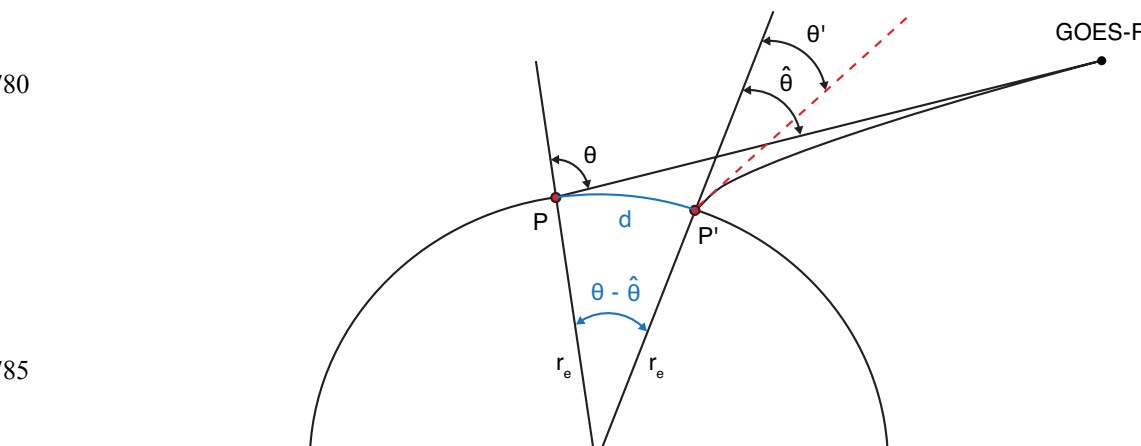


**Figure 5.** Terrestrial refraction geometry after Noerdlinger (1999). The distant observer (GOES-R) views point $P'$, which is displaced by distance $d$ and registered in the satellite image at point $P$. Noerdlinger (1999) provides analytical formulas that relate the known zenith angle $\theta$ of the unrefracted ray to $\hat{\theta}$ and $\theta'$, from which the horizontal displacement along the surface can be calculated using the local Earth curvature radius $r_e$ (= 6,371,000 m in the spherical model).







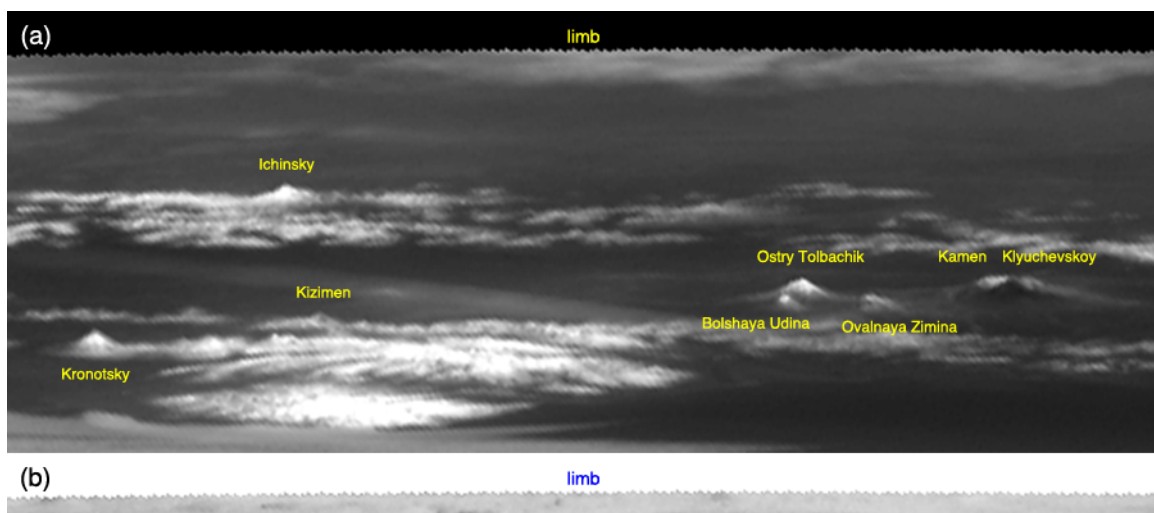



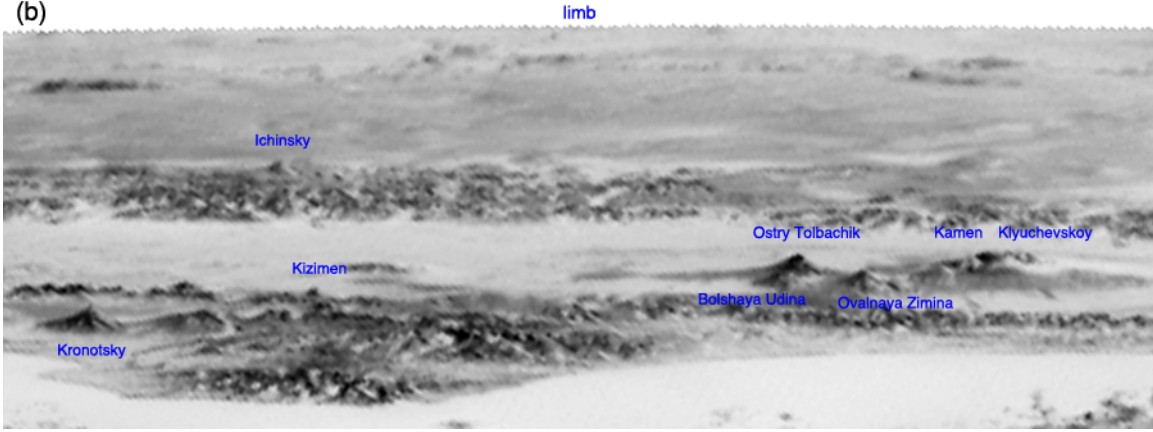

**Figure 6.** GOES-17 band 2 fixed grid image of central Kamchatka on **(a)** 13 June 2020 at 23:00 UTC and **(b)** 21 December 2019 at 23:50 UTC with the most prominent volcanic peaks labeled. The images were magnified by a factor of 3, rotated clockwise by 31° to ensure a horizontal limb, and an inverted black–white gradient map was applied to panel **(b)** to mimic a shaded relief effect.






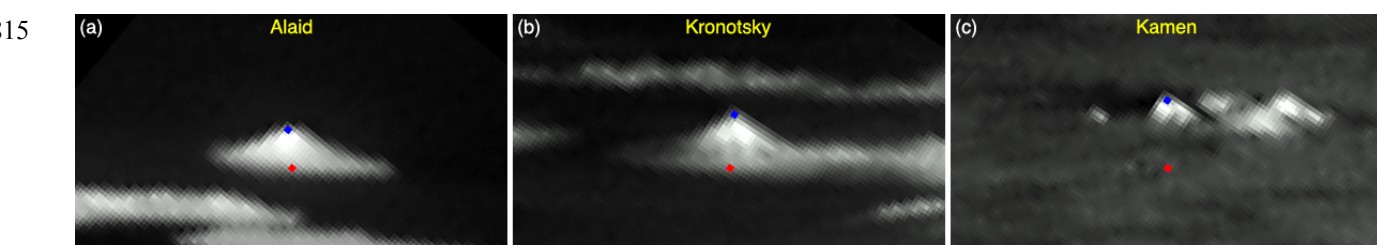


**Figure 7.** GOES-17 band 2 fixed grid image after 8x magnification of **(a)** Alaid (Atlasov Island, 2339 m) on 30 March 2020 at 00:00 UTC, **(b)** Kronotsky (3528 m) on 13 June 2020 at 23:00 UTC, and **(c)** Kamen (4619 m) on 8 April 2020 at 19:00 UTC. The image pixel corresponding to the geodetic latitude/longitude of the vent is marked by a red diamond. The visually identified peak with a 1-pixel radial refraction correction applied is marked by a blue diamond. The images were rotated
clockwise by the geodetic colatitude angle. In panel **(c)**, the tops of (left to right or south to north) Kamen, Ushkovskiy (3891 m), Klyuchevskoy (4835 m), and Krestovsky (4048 m) are seen peeking through the cloud layer.



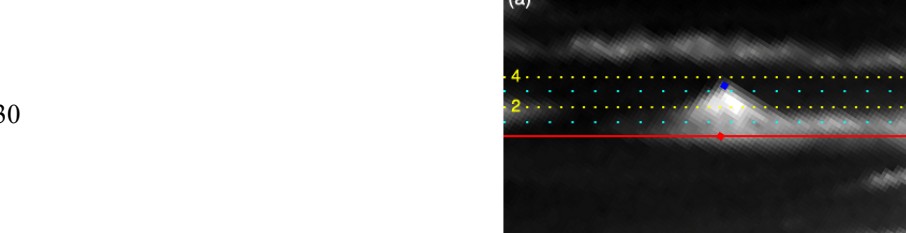

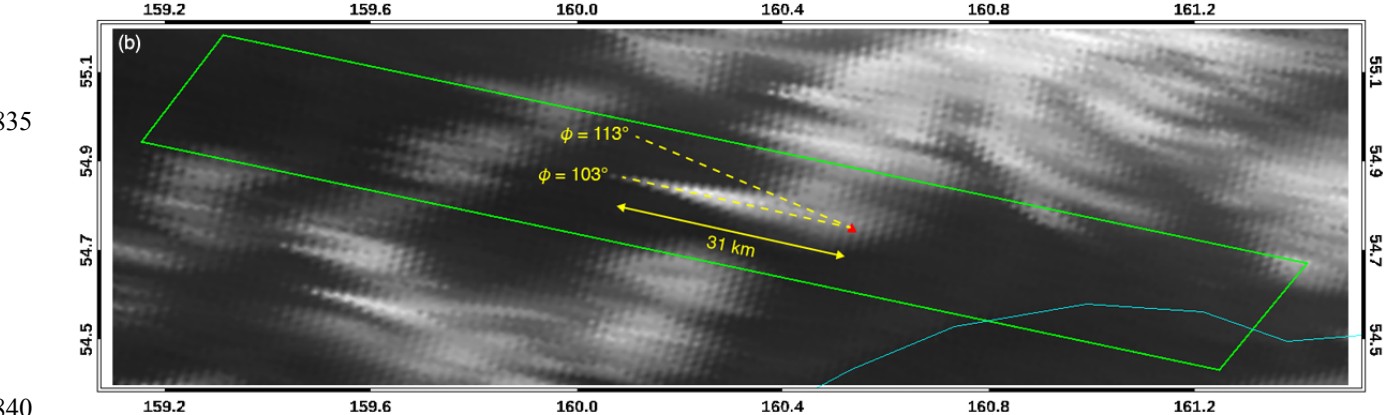

**Figure 8.** GOES-17 band 2 image of Kronotsky (3528 m) on 13 June 2020 at 23:00 UTC in **(a)** fixed grid projection (8x magnification) and **(b)** equirectangular (Plate-Carrée) projection. In panel **(a)**, the red and blue diamonds mark the volcano base and peak, respectively, and the horizontal lines are base-relative isoheights drawn at 1 km intervals (solid red: surface, cyan dotted: odd numbers, yellow dotted: even numbers). In panel **(b)**, the green quadrilateral bounds the area shown in panel **(a)**, the red triangle indicates the volcano base, and the cyan curve is the coastline. The true view azimuth ($\phi = 113°$), the apparent (distorted) azimuth ($\phi = 103°$), and the ellipsoid-projected distance between the base and peak locations are also indicated.





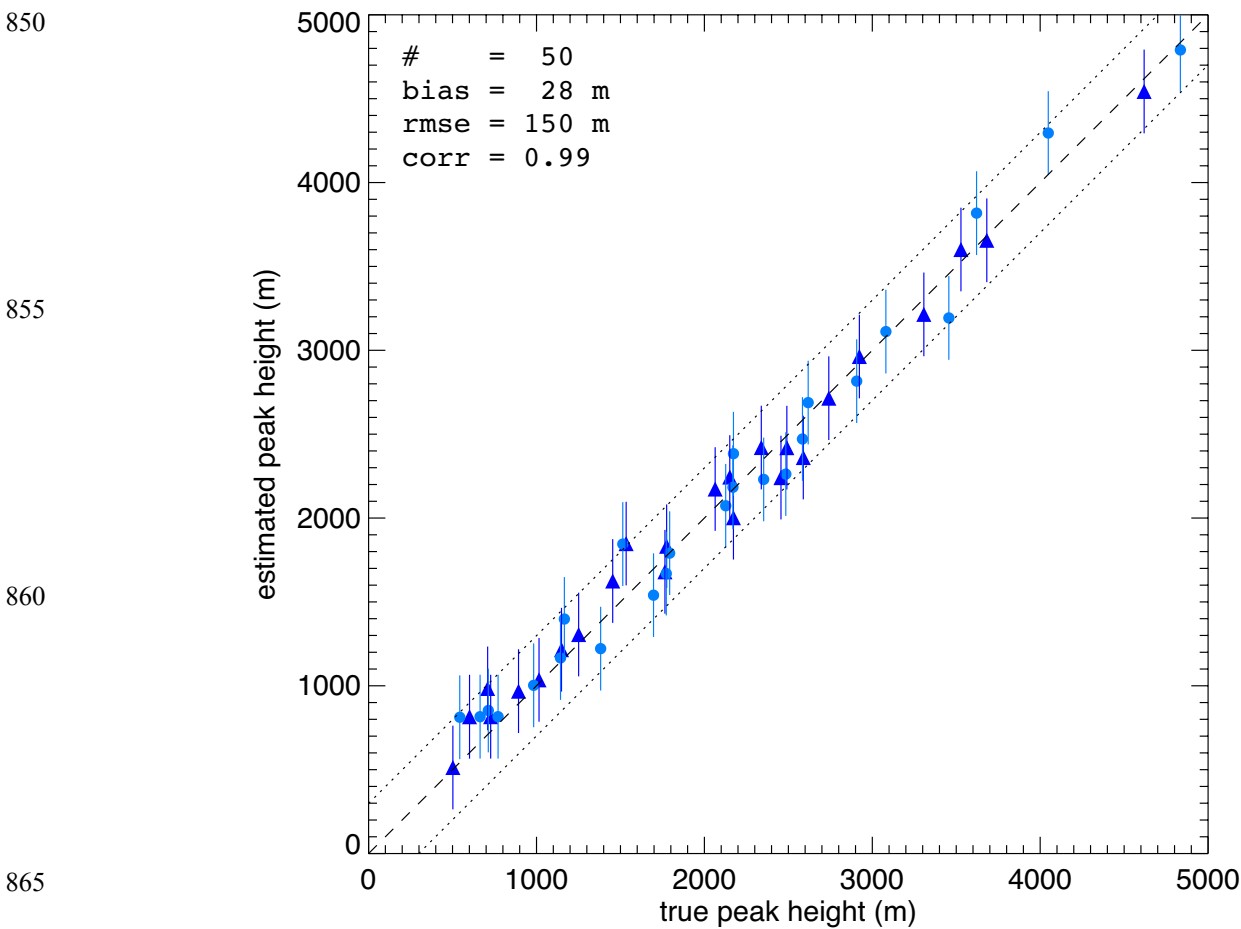

**Figure 9.** Peak height estimated by the side view method versus true peak height; two different colors and symbols are used only to help distinguish overlapping data points. The dashed line is the 1:1 line and the dotted lines mark ±2 × RMSE about the 1:1 line. The error bars on a data point represent the standard deviation of the nine height estimates corresponding to the visually identified peak location and its 8-pixel neighborhood.



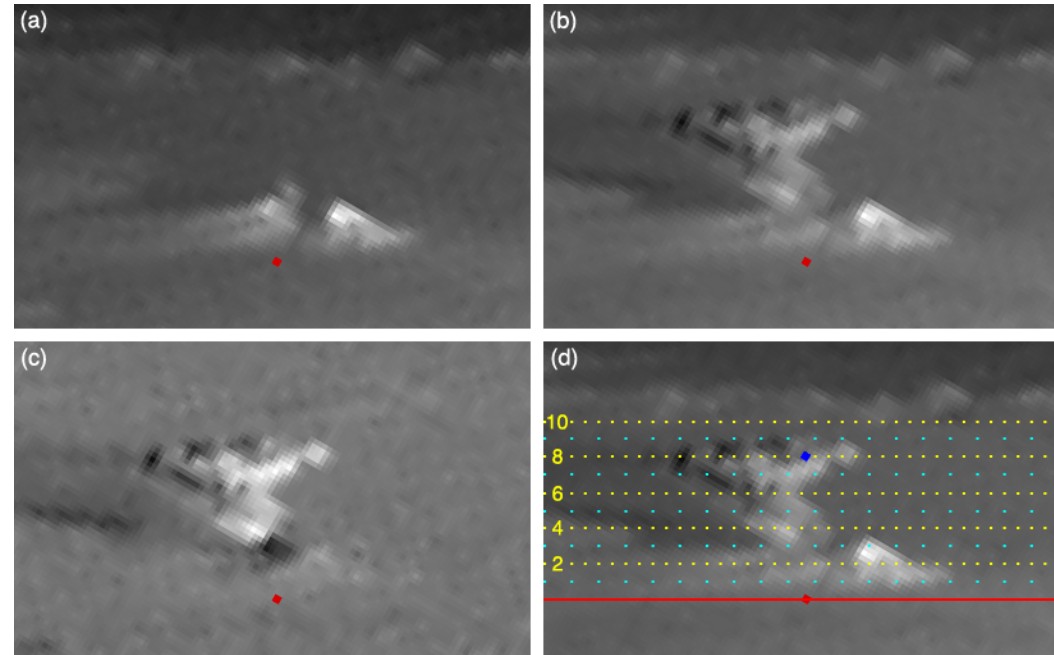

**Figure 10.** GOES-17 band 2 fixed grid image (8x magnification) of Young and Old Sheveluch (2589 m and 3307 m, respectively) on 8 April 2020 at **(a)** 19:00 UTC, **(b)** 19:10 UTC, **(c)** 19:10 UTC running-difference image, and **(d)** 19:10 UTC with base-relative isoheight lines drawn at 1 km intervals. The images were rotated clockwise by the geodetic colatitude angle. The red diamond marks the base of the active Young Sheveluch and the blue diamond indicates our best visual estimate plume top position above the vent.





895

900

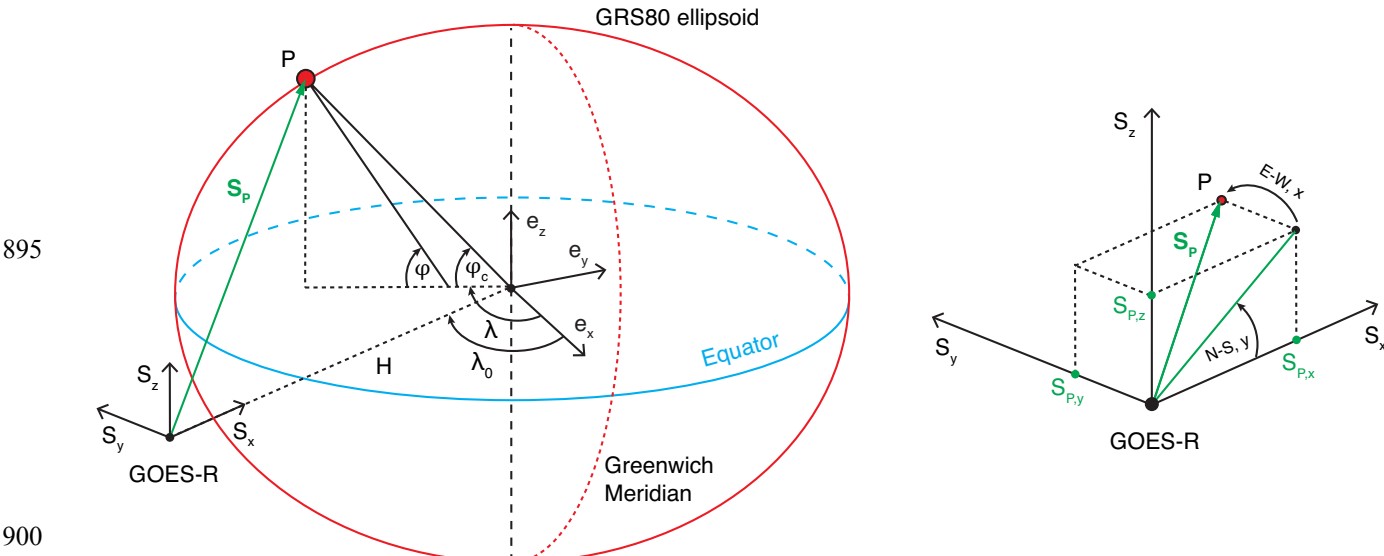

**Figure A1.** Coordinate systems used for ABI fixed grid navigation.