# Peer review of "Geometric estimation of volcanic eruption column height from GOES-R near-limb imagery – Part 1: Methodology"

_Atmospheric Chemistry and Physics, 2021_

## Author Response (AR1)

**Part 1**

We thank the Referees for their constructive comments. In the following, we give a point-by-point response to each of the issues raised.

**Referee #1**

*Row 40: just a comment to say that obviously the temperature profile can be obtained not only from a numerical forecast but also, for example, from a reliable measurement such as radiosounding if available in space and time concomitance with the eruption. Since we are here in the introduction of the paper, I would give a more general explanation to the BT method.*

We have modified the sentence to state that the temperature profile can be obtained from a radiosounding or a numerical forecast.

**Referee #2**

*The only thing I would like to address is the conclusion - at the moment it seems more like a summary to me, make it more concise - focus on your scientific contribution and a relation to part 2.*

We have made the Summary more concise by removing discussion not strictly pertinent to the presented methodology. We have also added a segue into Part 2.

*Section 3.2. What about the case when the volcanic cloud contains a lot of "topography"? Then measuring the length from the side considered only the height of the side of the cloud.*

We have added a sentence to the first paragraph of section 3.2, noting that for a significantly bumpy plume top the surface-measured shadow length leads to height underestimation, because the highest plume point might cast its shadow on the lower and wider parts of the plume itself.

*Your list of references is good, but considering your geometric approach, I would suggest adding:*
*https://www.mdpi.com/2072-4292/11/7/785*
*https://www.sciencedirect.com/science/article/abs/pii/S0034425718300737*

We have included the de Michele et al. (2019) and Zakšek et al. (2018) references in section 3.4, as suggested.